# Effects of the IR Drop on the Electrochemical Corrosion of X80 Pipeline Steel in Different Solutions

**Yunlong Bai** [1,2,†], **Jin Xu** [1,2,3,*,†], **Boxin Wei** [1,2] and **Cheng Sun** [1,2,3,*]

1   School of Materials Science and Engineering, University of Science and Technology of China, Shenyang 110016, China
2   Institute of Metal Research, Chinese Academy of Sciences, Shenyang 110016, China
3   Liaoning Shenyang Soil and Atmosphere Material Corrosion National Observation and Research Station, Shenyang 110016, China
*   Correspondence: xujin@imr.ac.cn (J.X.); chengsun@imr.ac.cn (C.S.)
†   These authors contributed equally to this work.

**Abstract:** Polarization curves are popularly used in many investigations; however, some wrong results are obtained when inexperienced researchers do not deeply understand electrochemical processes. In this study, the effects of the IR drop (Deviation due to I (current) and R (resistance)) and scanning direction on polarization curves were investigated. Three different test methods were used to test the polarization curve of X80 pipeline steel in five kinds of solutions. The different scanning directions led to differences in the cathodic polarization curves. In addition, the effect of IR drop on the Tafel curve in acidic solutions is significantly greater than in neutral and alkaline solutions. There is a large effect of the IR drop on shapes and fitting results of polarization curves in acidic solutions, and the IR drop must be considered for the fitting of curves. Scanning direction has an influence on the shape of polarization curves when a layer of corrosion products is formed on the steel surface, and a novel two-electrode coupon was introduced to solve this problem.

**Keywords:** polarization curve; IR drop; electrochemical corrosion; X80 steel





## 1. Introduction

Corrosion of a metal is usually evaluated using a gravimetric method, e.g., a weight-loss method; however, there are some disadvantages, for example, (1) it is time-consuming; (2) it can only give corrosion rates calculated by weight variations and not provide any other data; (3) it cannot accurately measure the corrosion rate of metals with very low rates. With the development of electrochemical techniques, polarization curves have become popular in many fields, such as the corrosion of metals [1–3], the corrosion of reinforced concrete [4], corrosion inhibitors [5,6], coatings [7], fuel cells [8], and so on. Electrochemical techniques have been shown to be very useful for discussing mechanisms in laboratory studies and monitoring purposes in field investigations.

Potentiodynamic polarization is a useful technique due to the reason that it can ascertain some electrochemical parameters, e.g., corrosion potentials, corrosion current densities, and Tafel slopes of cathodes and anodes, from polarization curves by corresponding fitting software, and further confirm the corrosion rates and corrosion processes of metals. The polarization curve method also has some limitations, e.g., the resistance of the solution (usually called the IR drop) and the selection of the Tafel portions of anodic and cathodic curves. Only after correct data of the polarization curve are selected, can the electrochemical parameters be fitted accurately.

The contribution of the IR drop to the electrode potential cannot be neglected when the current intensity is big enough, especially for high-polarization regions. The IR drop can also influence the apparent value of the polarization resistance dramatically and mislead the evaluation of the real corrosion state, causing a misinterpretation of selected experimental

data [9]. Although some electrochemical workstations have a function of IR compensation, this function is not always used by inexperienced researchers.

In the early period, polarization curves were measured by a potentiostat controlled by hands [10], and a cathodic curve was firstly scanned from the open-circuit potential (OCP) of a metal in the negative direction before an anodic curve was scanned from the OCP in the positive direction once the OCP was re-stable [11]. With the development of computer technologies, the human-controlled electrochemical instruments were gradually replaced by auto-controlled potentiostats/galvanostats. At the same time, some fitting programs were developed to obtain electrochemical parameters, such as corrosion potentials, corrosion currents, and the Tafel slopes of cathodes and anodes. However, because some novice researchers directly measured polarization curves using programs designed by computers, the scanning methods also began to change, i.e., the polarization curve is scanned from the cathodic to anodic potentials for convenience, which might lead to some problems in some solutions, although this method is workable in other systems. In different corrosion media, the corrosion state and corrosion product layer on the surface of the working electrode (WE) are different. The surface state of steel and micro-environments at the metal–solution interface can be changed by applying the potentiodynamic polarization. Additionally, the scanning methods of polarization curves can influence the surface conditions of the WE. Thus, the fitting results under different measuring conditions may be different from the actual ones [12–15]. The deviated data and curves can cause wrong-fitting results, which will draw misleading conclusions for readers and other researchers.

The object of this study is to gain an insight into more accurately obtaining the data of polarization curves with and without IR drops by comparing three methods in five solutions, and give some guidance to inexperienced scientists who want to devote themselves to corrosive and electrochemical fields.

## 2. Materials and Methods

### 2.1. Coupon Preparations

X80 pipeline steel (Baowu Iron and Steel Group Co., Wuhan, China) was used. It has a nominal composition ($w/w$ %) of C 0.070, Mn 1.820, P 0.007, S 0.023, Si 0.190, Mo 0.010, Ni 0.170, Cr 0.026, Cu 0.020, V 0.002, Nb 0.056, Ti 0.012, Al 0.028, and Fe balance. It was cut into cubic coupons (10 mm × 5 mm × 3 mm), and two coupons were mounted in epoxy resin as a working electrode with a working surface area of 1 cm$^2$, as shown in Figure 1. The coupons were sequentially ground with a series of SiC papers (150, 240, 400, and 600 grit). They were then washed sequentially with deionized water, acetone, and 100% ethanol (Sinopharm, Shanghai, China) to clean and degrease the surfaces. The electrodes were dried and kept in a desiccator.

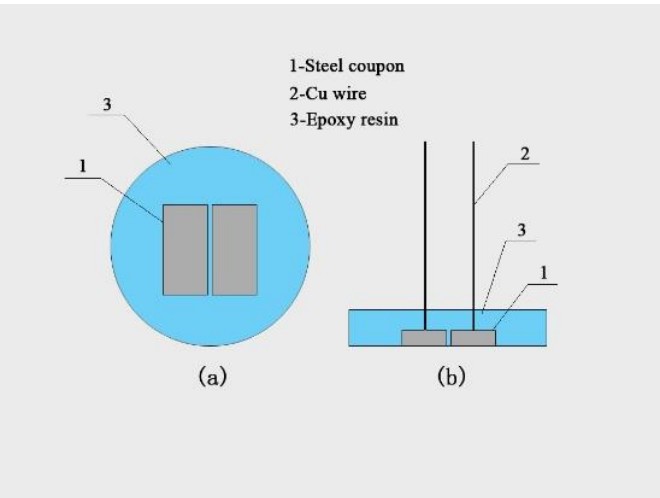

**Figure 1.** Schematic diagram of the steel coupon with main (**a**) and side (**b**) views.

## 2.2. Testing Solutions

Five solutions, including HCl (Sinopharm, Shanghai, China), $H_2SO_4$ (Sinopharm, Shanghai, China), NaCl (Sinopharm, Shanghai, China), $Na_2SO_4$ (Sinopharm, Shanghai, China), and NaOH (Sinopharm, Shanghai, China) were used to measure the polarization curves of X80 steel, and the concentrations of these solutions were all 1 mol/L. The reagents used in this study are all AR grade.

## 2.3. Solution Resistance Drop

There is solution resistance between a working electrode and a reference electrode. An IR drop will be produced when a current/potential is applied to the working electrode. Generally, this IR drop is not always considered for the measurement and datum fitting of polarization curves. However, the contribution of the IR drop to the potential of a working electrode cannot be neglected when the current density or the solution resistance is big enough, especially when the polarization curve enters into high-polarization regions. The generation of an IR drop causes the deviation of the measuring potential from the actual polarization potential applied to the working electrode, which leads to mistaken fitting results of polarization curves, e.g., corrosion current density, the Tafel slopes of anode and cathode, or misleading the evaluation of corrosion process.

The true polarization potential of an electrode can be calculated as follows:

$$|\Delta E| = |\Delta E_m| - |i| R_{sol} \tag{1}$$

where $\Delta E$ is the actual value of the polarization potential of a working electrode; $\Delta E_m$ is the measuring value of the electrode potential; $i$ is the measuring value of the current density, and $R_{sol}$ is the solution resistance between the working electrode and the reference electrode.

Several methods were used to eliminate the IR drop, for example, the method of two reference electrodes [16], the method of stationary equi-space potential [17], and the electrochemical impedance spectroscopy (EIS) method [18]. Compared with other methods, the EIS method is easily used to measure the solution resistance and has almost no effects on the surface of a coupon.

## 2.4. High-Polarization Region of Tafel Extrapolation

For an electrode reaction described as the following Reaction (2), the current density is often related exponentially to the overpotential [19].

$$O + ne^- \leftrightarrow R \tag{2}$$

where $O$ and $R$ represent an oxidation state and a reduction state, respectively.

The equation of the current density and overpotential can be expressed as:

$$i = i_0 e^{\frac{\alpha \eta F}{nRT}} \tag{3}$$

where $i$ is the current density; $i_0$ is the exchange current density; $\alpha$ is the transfer coefficient, and $\eta$ is the overpotential.

The current densities of the forward reaction and reverse reaction of Reaction (2) can be expressed as the Equations (4) and (5):

$$i_{For} = i_0 e^{\frac{\alpha \eta F}{nRT}} \tag{4}$$

$$i_{Rev} = i_0 e^{\frac{(1-\alpha)\eta F}{nRT}} \tag{5}$$

where $i_{For}$ and $i_{Rev}$ are the forward and reverse current densities, respectively.

The overall current density of Reaction (2) is:

$$i = i_0 \left[ e^{\frac{\alpha \eta F}{nRT}} - e^{\frac{(1-\alpha)\eta F}{nRT}} \right] \tag{6}$$

As is well known, a corrosion reaction is divided into at least an anodic reaction and a cathodic reaction. According to Reaction (2), the anodic reaction means that the reverse reaction rate is much larger than the forward reaction rate, namely $i_{Rev} \gg i_{For}$, and thus the current density of the anodic reaction from Equation (6) is:

$$|i_a| = i_{corr} e^{\frac{(1-\alpha_a)|\Delta E|F}{nRT}} = i_{corr} e^{\frac{|\Delta E|F}{\beta_a}} \tag{7}$$

Similarly, the current density of the cathodic reaction is as follows,

$$|i_c| = i_{corr} e^{\frac{-\alpha_c|\Delta E|F}{nRT}} = i_{corr} e^{\frac{-|\Delta E|F}{\beta_c}} \tag{8}$$

where the subscripts $a$ and $c$ represent anode and cathode, respectively.

When the polarization curve enters into the high-polarization region (namely the Tafel region), it is generally accepted that the contribution of the anodic reaction is more than 1% of the total reaction in the anodic Tafel region, and vice versa. Therefore, there is a relationship between anodic and cathodic current densities.

$$\frac{e^{\frac{|\Delta E|}{\beta_a}}}{e^{\frac{-|\Delta E|}{\beta_c}}} = e^{|\Delta E|\left(\frac{1}{\beta_a} + \frac{1}{\beta_c}\right)} \geq 0.01 \tag{9}$$

$$|\Delta E| \geq \frac{4.605\beta_a\beta_c}{\beta_a + \beta_c} = \frac{2b_ab_c}{b_a + b_c} \tag{10}$$

For iron and steel, $b_a$ is usually from 40 to 80 mV/dec and $b_c$ is about 120 mV/dec under actively anodic dissolution in an acid solution [20]. As a result, the polarization curve enters into the high-polarization region when the $|\Delta E|$ value reaches 96 mV.

In this study, the data in the range from 100 to 150 mV were selected for the Tafel linear fitting using the OriginPro software version 8.5 (OriginLab Corporation, Northampton, MA, USA), obtaining the fitting results of polarization curves.

*2.5. Electrochemical Measurements*

A conventional three-electrode glass cell (Gaoss Union, Tianjin, China) was used to measure electrochemical impedance spectra (EIS) and polarization curves using the Gamry electrochemical workstation (600+, Gamry, Warminster, PA, USA). It had an X80-pipeline-steel working electrode, a counter electrode of platinum foil with an area of 4 cm$^2$, and a calomel (saturated KCl, Shanghai INESA Scientific Instrument Co., LTD, Shanghai, China) reference electrode which was connected to the cell by a salt bridge.

An EIS measurement was used to obtain the solution resistance between the working electrode and the reference electrode to calculate the potential drops (IR drop) induced by the solution resistance and conducted at the open circuit potential (OCP) with a 10 mV sinusoidal signal over the frequency range of 100 kHz to 100 Hz. The EIS data were fitted by ZSimpWin software (version 3.21, Echem Software, Ann Arbor, MI, USA).

Polarization curves were measured according to the following three methods:

(1) Measurement 1 (M1)

Two coupons (Figure 1) were connected electrically during the whole experiment, simulating a traditional working electrode with a size of 10 mm × 10 mm. The polarization curve was scanned from −300.0 mV to +300.0 mV vs. OCP after the OCP potential was stable.

(2) Measurement 2 (M2)

Two coupons (Figure 1) were also connected electrically during the whole experiment as a working electrode. First, the polarization curve was scanned from the OCP to $-300.0$ mV (vs. OCP), and then from the OCP to $+300.0$ mV (vs. OCP) after the OCP potential was stable again.

(3) Measurement 3 (M3)

Two coupons (Figure 1) were also connected electrically during the whole experiment as a working electrode except for the measuring periods. When the electrochemical measurements were conducted, these two electrically connected coupons (Figure 1) were divided into two working electrodes, one for the measurement of the cathodic curve and the other for the anodic curve. First, one half-electrode was used to measure the cathodic curve range from the OCP to $-300.0$ mV vs. OCP, while the other half-electrode was used to measure the anodic curve range from the OCP to $+300.0$ mV (vs. OCP).

Cao [15] has also concluded that the polarization curves of iron and steel reach Tafel linear regions when the $|\Delta E|$ value is more than 100 mV under an acid-dissolution condition. Thus, the data, $|\Delta E|$ from 100.0 mV to 150.0 mV (vs. $E$ without IR drop), measured using the three methods mentioned above, were selected for Tafel linear fitting, and were fitted using linear regression with OriginPro software.

## 3. Results

Figure 2 shows the comparison of the polarization curves measured by three measuring methods in five solutions including HCl, $H_2SO_4$, NaCl, $Na_2SO_4$, and NaOH. For acidic and alkaline solutions, Figure 2a,b, the shapes of the three polarization curves are almost the same in appearance, while for neutral and alkaline solutions, there are huge differences between the three polarization curves, especially for the cathodic curves. It can be seen from Figure 2c–e that horizontal lines (green circles), just like tails, are observed in the cathodic regions of the polarization curves measured by M1 compared with those by M2 and M3. The maximum values of cathodic current densities are 131.3 $\mu A \cdot cm^{-2}$ (M1), 48.3 $\mu A \cdot cm^{-2}$ (M2), and 39.5 $\mu A \cdot cm^{-2}$ (M3) for the NaCl solution, 465.4 $\mu A \cdot cm^{-2}$ (M1), 63.4 $\mu A \cdot cm^{-2}$ (M2), and 61.2 $\mu A \cdot cm^{-2}$ (M3) for the $Na_2SO_4$ solution, and 10.8 $\mu A \cdot cm^{-2}$ (M1), 8.0 $\mu A \cdot cm^{-2}$ (M2), and 7.4 $\mu A \cdot cm^{-2}$ (M3) for the NaOH solution, respectively. There are big differences between the current densities for M1 and those of M2 or M3, especially for the NaCl and $Na_2SO_4$ solutions. Too large cathodic current density must influence the surface states of coupons and further change the cathodic and anodic processes. The anodic polarization curve with an active-passive region is observed in Figure 2d, which is very different from those measured by the other two methods because of the highest value of the initial cathodic current. Similarly, the cathodic curve of M1 also shows some differences compared with the curves by M2 and M3.

Table 1 gives the maximum anodic and cathodic potentials of the Tafel curves of X80 steel coupons using three methods in five solutions. As described in paragraph 2.3, the maximum potentials of anodic and cathodic scanning are 300 mV and $-300$ mV, respectively. However, the anodic and cathodic scanning ranges of the curves show, to some extent, an asymmetry for the M1 measurement in five solutions, as shown in Table 2, and almost none for the M2 and M3 measurements, which may be related to the properties of solutions. It can also be found that the potential deviations reach nearly 40 mV in neutral solutions, which are larger than those in alkaline and acidic solutions. It can be seen from Figure S1 that corrosion product layers are obviously observed on the steel coupons in neutral solutions, but not in alkaline and acidic solutions. It can be inferred that the formation of corrosion products is one reason causing the shifts in the scanning potentials.

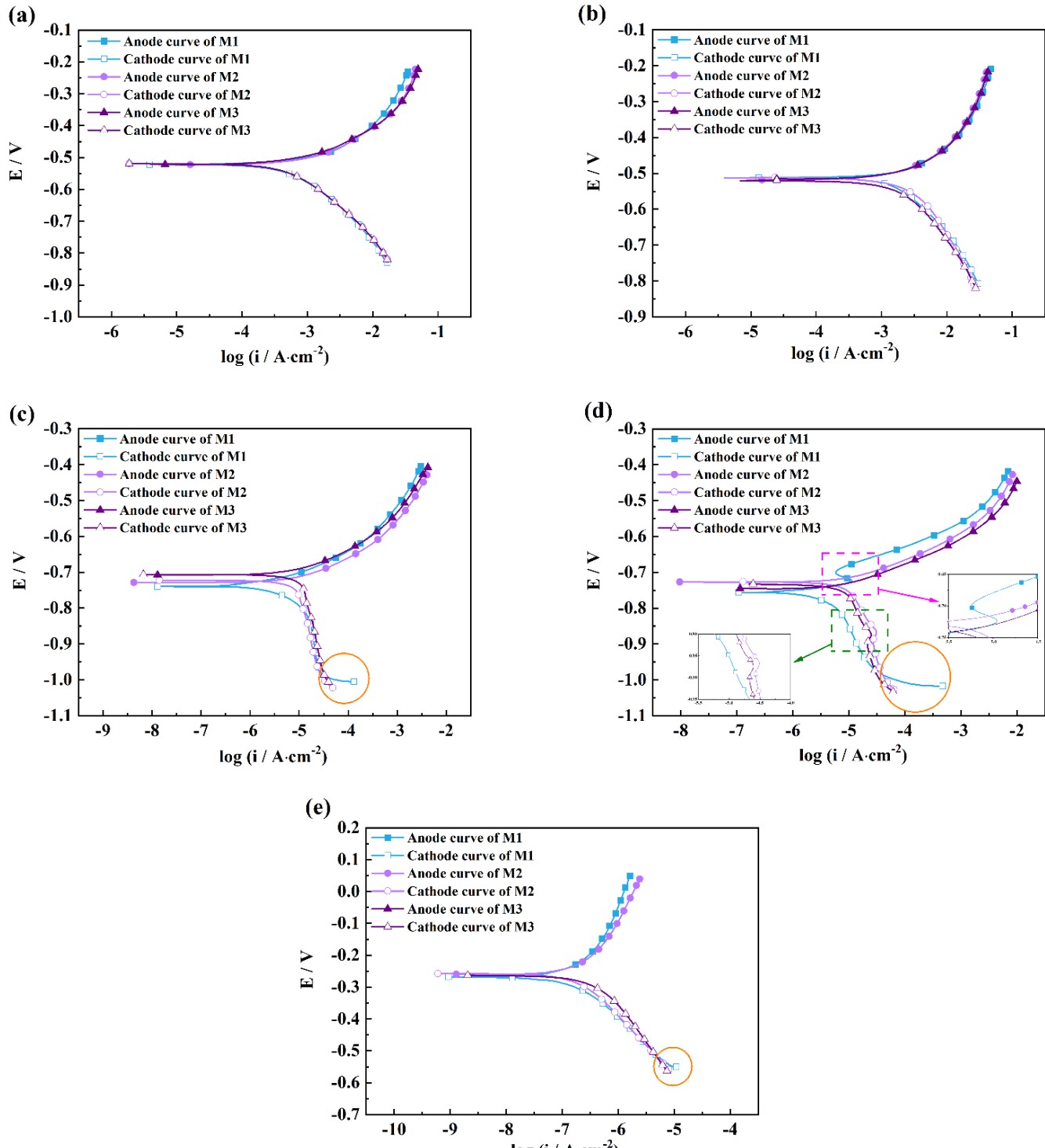

**Figure 2.** Comparison of the strong-polarization curves measured by three measuring methods in five solutions of HCl (**a**), H$_2$SO$_4$ (**b**), NaCl (**c**), Na$_2$SO$_4$ (**d**), and NaOH (**e**). (Skip points were 80 for clarity.)

**Table 1.** Maximum anodic and cathodic current densities of Tafel curves using three methods in five solutions.

| Solution | | HCl | | | H$_2$SO$_4$ | | | NaCl | | | Na$_2$SO$_4$ | | | NaOH | | |
|---|---|---|---|---|---|---|---|---|---|---|---|---|---|---|---|---|
| Measuring method | | M1 | M2 | M3 | M1 | M2 | M3 | M1 | M2 | M3 | M1 | M2 | M3 | M1 | M2 | M3 |
| Scanning maximum value | Anode | 282 | 299 | 301 | 296 | 300 | 299 | 334 | 300 | 300 | 339 | 300 | 299 | 317 | 299 | 299 |
| | Cathode | 308 | 301 | 299 | 303 | 300 | 299 | 266 | 300 | 300 | 261 | 300 | 300 | 283 | 301 | 300 |

**Table 2.** Solution resistance fitted by EIS.

| | HCl $\Omega \cdot \text{cm}^2$ | H$_2$SO$_4$ $\Omega \cdot \text{cm}^2$ | NaCl $\Omega \cdot \text{cm}^2$ | Na$_2$SO$_4$ $\Omega \cdot \text{cm}^2$ | NaOH $\Omega \cdot \text{cm}^2$ |
|---|---|---|---|---|---|
| M1 | 5.315 | 4.744 | 27.71 | 18.92 | 12.14 |
| M2 | 3.237 | 5.425 | 19.09 | 14.23 | 11.29 |
| M3-anode | 3.182 | 5.123 | 15.18 | 12.02 | 10.06 |
| M3-cathode | 4.145 | 4.579 | 11.56 | 13.77 | 9.97 |

## 4. Discussion

### 4.1. Effect of Solution Resistance

As is well known, the IR drop will be produced when a current/potential is applied to the working electrode. Generally, this IR drop is not considered for the treatment of a polarization curve. However, the contribution of the IR drop to the potential of the working electrode cannot be neglected when the current density applied or the solution resistance is big enough, especially for high-polarization regions. The generation of the IR drop causes the deviation of the measuring potential from the actual polarization potential applied to the working electrode, which leads to mistaken calculations of the fitting results of the polarization curve, e.g., corrosion current density, the Tafel slopes of anode and cathode, and misleading the evaluation of a corrosion process.

When a polarization curve is performed, as shown in Figure 3, the overpotential is applied from the outside between the points A and C. However, the real polarization potential, which is the potential difference between B and C and determines the electrode process, usually differs from the measured one.

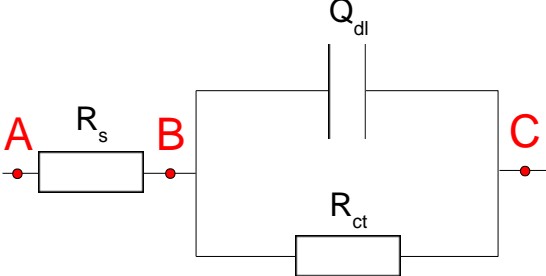

**Figure 3.** The equivalent electric circuit of the metal–solution interface.

In this case, there exists a relation as the following Equation (11),

$$E_{\text{R}} = E_{\text{M}} - i \cdot R_{\text{S}} \tag{11}$$

where $E_{\text{R}}$ is the real value of the potential difference between B and C; $E_{\text{M}}$ is the measuring value of the potential difference between A and C; $R_{\text{S}}$ is the solution resistance, and $i$ is the measuring current density. Thus, Equation (11) shows that the existence of the ohmic drop influences not only the shape of the polarization curve but also the values of the electrochemical parameters if the $|i| R_{\text{S}}$ value is large enough. To accurately evaluate the slopes of the Tafel straight lines, it is necessary to remove the contribution of the IR drop to the electrode potential. The EIS method is usually used to measure the solution

resistance in corrosive fields. Table 2 gives the solution resistances of five solutions fitted using the EIS data.

Figures 4–7 show the comparison plots of polarization curves with and without an IR drop under three measurements and linear fitting plots of Tafel regions, and the calculated $E_{corr}$, $i_{corr}$, $b_a$, and $b_c$ values are listed in Tables 3–6 together with the corresponding fitting Adj. $R^2$ values [21–25].

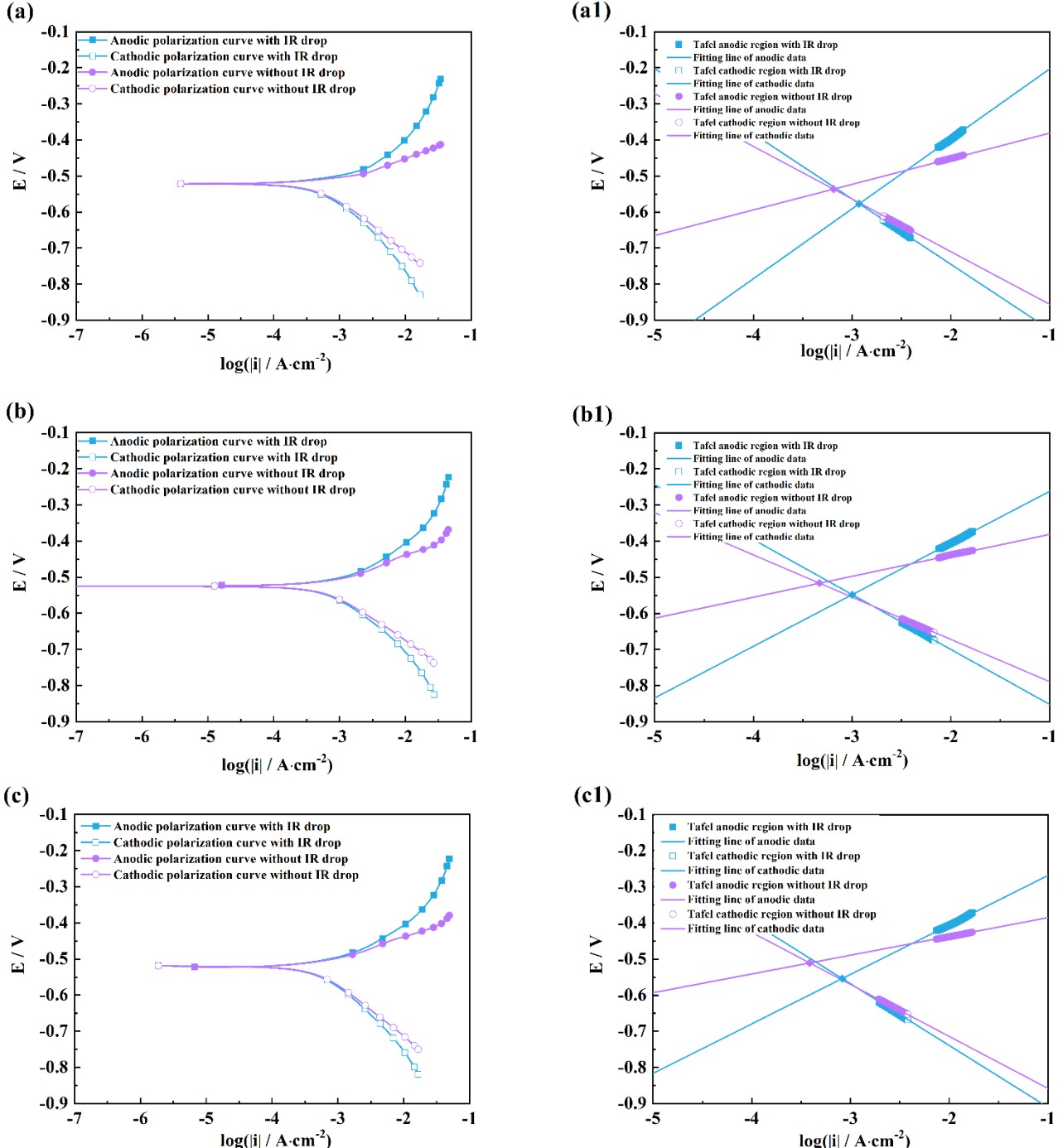

**Figure 4.** Comparison of polarization curves with and without an IR drop of M1 (**a**), M2 (**b**), M3 (**c**), and linear fitting plots of the Tafel regions of M1 (**a1**), M2 (**b1**), M3 (**c1**) in HCl solutions after 2 h exposure.

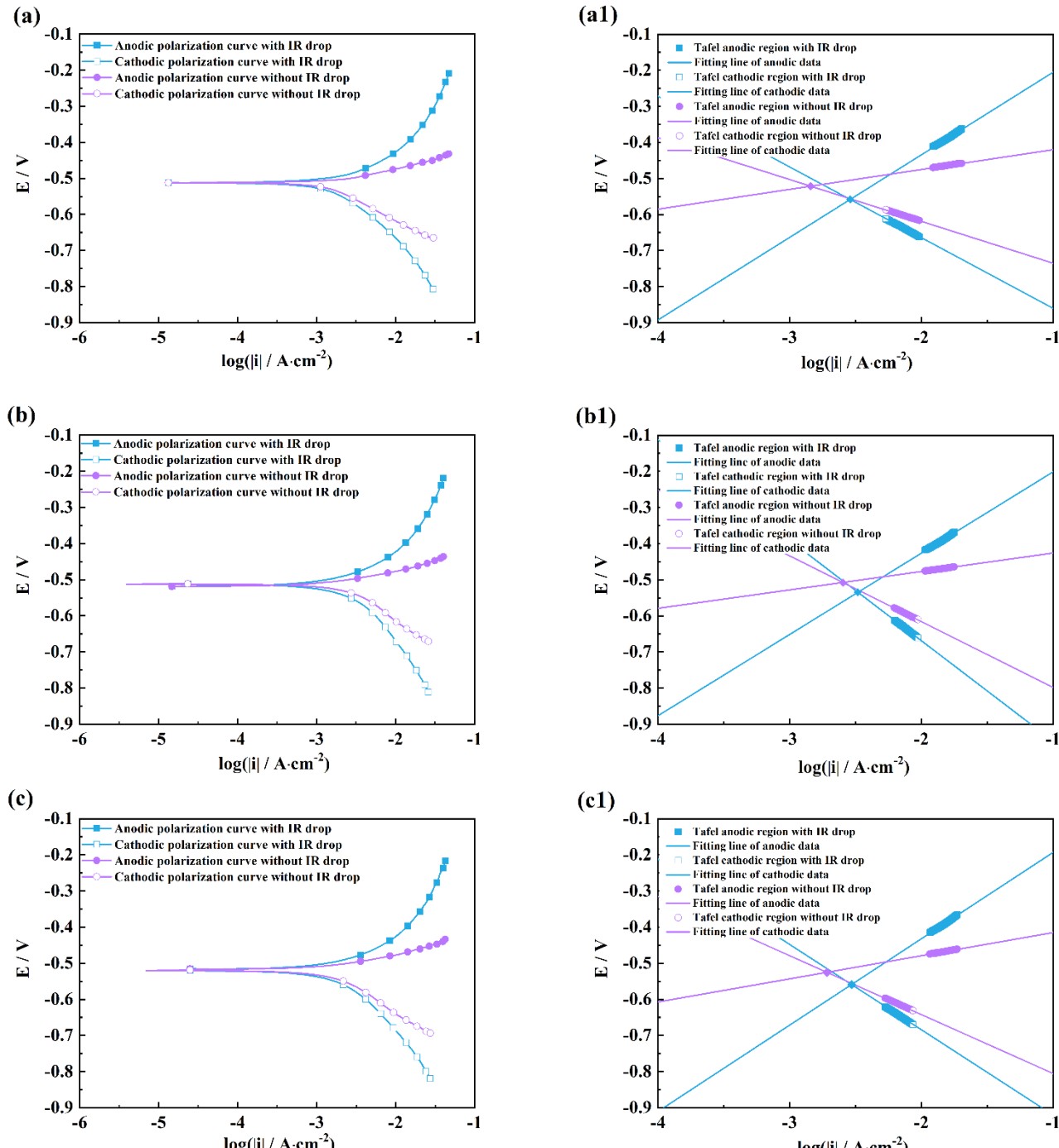

**Figure 5.** Comparison of polarization curves with and without an IR drop of M1 (**a**), M2 (**b**), M3 (**c**) and linear fitting plots of the Tafel regions of M1 (**a1**), M2 (**b1**), M3 (**c1**) in $H_2SO_4$ solutions after 2 h exposure.

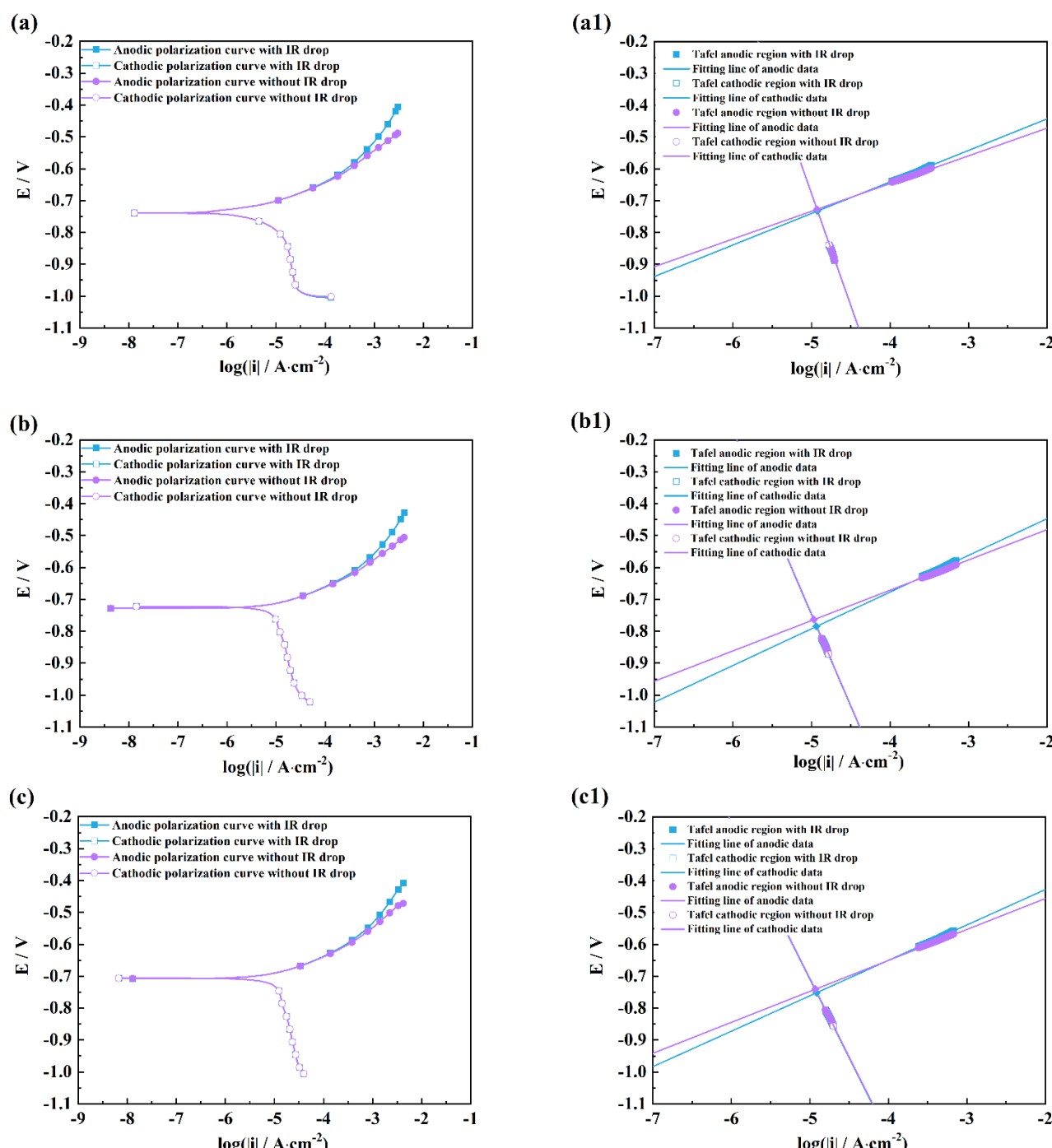

**Figure 6.** Comparison of polarization curves with and without an IR drop of M1 (**a**), M2 (**b**), M3 (**c**) and linear fitting plots of the Tafel regions of M1 (**a1**), M2 (**b1**), M3 (**c1**) in NaCl solutions after 2 h exposure.

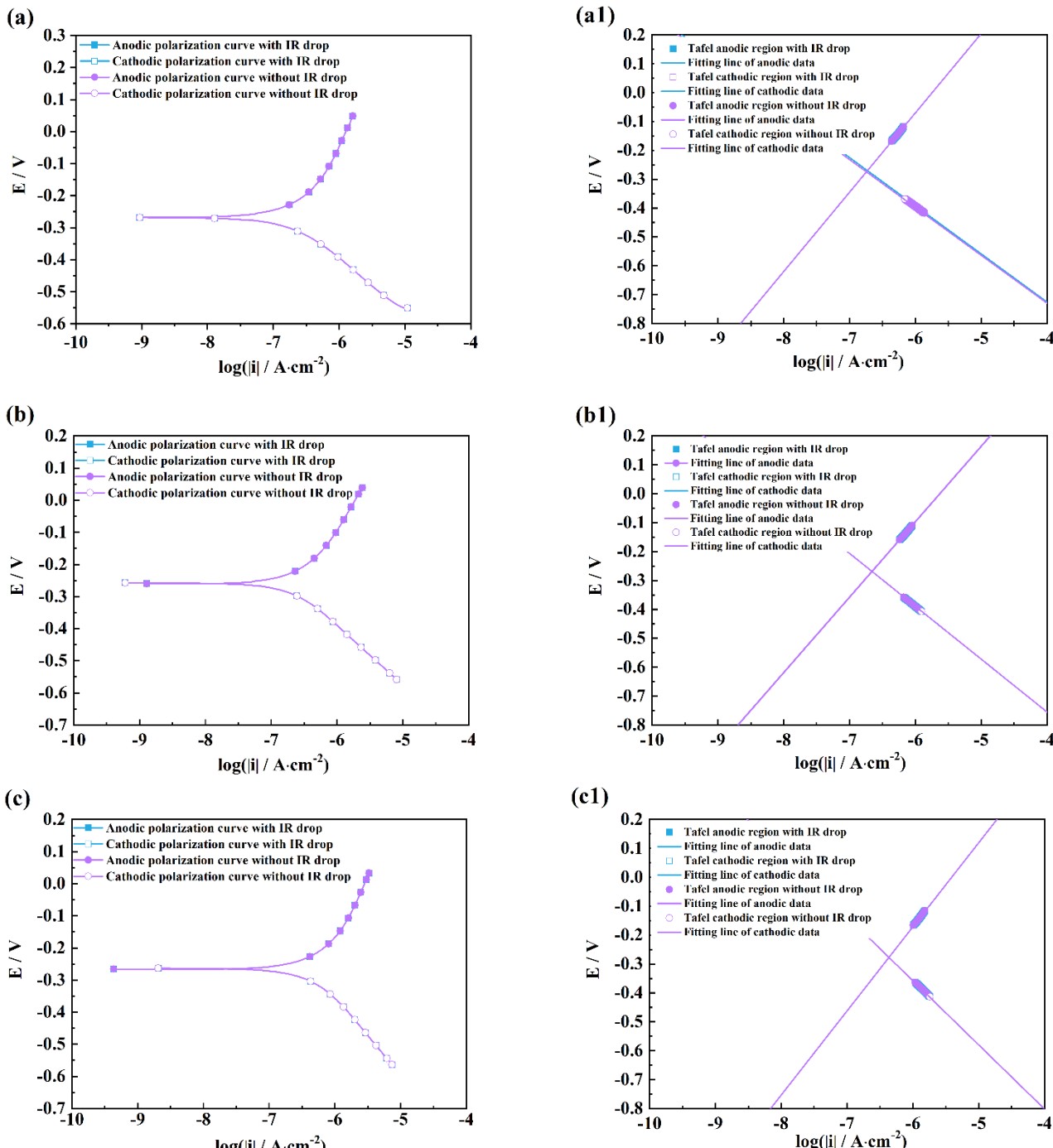

**Figure 7.** Comparison of polarization curves with and without an IR drop of M1 (**a**), M2 (**b**), M3 (**c**) and linear fitting plots of the Tafel regions of M1 (**a1**), M2 (**b1**), M3 (**c1**) in NaOH solutions after 2 h exposure.

**Table 3.** Fitting results of polarization curves with and without an IR drop by Tafel extrapolation under three measurements in HCl solutions after exposure for 2 h.

| Condtion | IR Drop | $E_{corr}$ mV | $i_{corr}$ μA·cm$^{-2}$ | $b_a$ mV·dec$^{-1}$ | $b_c$ mV·dec$^{-1}$ | Adj. R$^2$ [(1)] | |
|---|---|---|---|---|---|---|---|
| | | | | | | $b_a$ | $b_c$ |
| M1 | With | $-577.0 \pm 10.9$ | $1180 \pm 103$ | $194 \pm 33$ | $-181 \pm 21$ | 0.9972 | 0.9993 |
| | Without | $-536.4 \pm 9.1$ | $647 \pm 43$ | $71 \pm 6$ | $-146 \pm 13$ | 0.9995 | 0.9998 |
| M2 | With | $-548.0 \pm 8.9$ | $1000 \pm 55$ | $143 \pm 13$ | $-152 \pm 15$ | 0.9984 | 0.9989 |
| | Without | $-516.2 \pm 9.1$ | $466 \pm 29$ | $58 \pm 7$ | $-117 \pm 12$ | 0.9957 | 0.9997 |
| M3 | With | $-553.6 \pm 10.6$ | $823 \pm 73$ | $137 \pm 15$ | $-171 \pm 18$ | 0.9941 | 0.9997 |
| | Without | $-510.5 \pm 10.7$ | $386 \pm 28$ | $52 \pm 6$ | $-144 \pm 10$ | 0.9986 | 0.9999 |

[(1)] Adjusted R-squared of linear fitting E and log($|i|$) in the Tafel region of polarization curves.

**Table 4.** Fitting results of polarization curves with and without an IR drop by Tafel extrapolation under three measurements in H$_2$SO$_4$ solutions after exposure for 2 h.

| Condtion | IR Drop | $E_{corr}$ mV | $i_{corr}$ μA·cm$^{-2}$ | $b_a$ mV·dec$^{-1}$ | $b_c$ mV·dec$^{-1}$ | Adj. R$^2$ | |
|---|---|---|---|---|---|---|---|
| | | | | | | $b_a$ | $b_c$ |
| M1 | With | $-557.9 \pm 9.2$ | $2876 \pm 229$ | $229 \pm 23$ | $-196 \pm 20$ | 0.9975 | 0.9991 |
| | Without | $-521.3 \pm 9.9$ | $1438 \pm 141$ | $55 \pm 5$ | $-116 \pm 11$ | 0.9961 | 0.9996 |
| M2 | With | $-534.7 \pm 9.7$ | $3288 \pm 303$ | $225 \pm 21$ | $-278 \pm 24$ | 0.9961 | 0.9996 |
| | Without | $-507.4 \pm 9.8$ | $2532 \pm 217$ | $51 \pm 43$ | $-182 \pm 13$ | 0.9952 | 0.9991 |
| M3 | With | $-558.8 \pm 9.7$ | $2948 \pm 271$ | $239 \pm 33$ | $-236 \pm 21$ | 0.9958 | 0.9995 |
| | Without | $-525.0 \pm 10.5$ | $1914 \pm 163$ | $64 \pm 4$ | $-163 \pm 13$ | 0.9936 | 0.9997 |

**Table 5.** Fitting results of polarization curves with and without an IR drop by Tafel extrapolation under three measurements in NaCl solutions after exposure for 2 h.

| Condtion | IR Drop | $E_{corr}$ mV | $i_{corr}$ μA·cm$^{-2}$ | $b_a$ mV·dec$^{-1}$ | $b_c$ mV·dec$^{-1}$ | Adj. R$^2$ | |
|---|---|---|---|---|---|---|---|
| | | | | | | $b_a$ | $b_c$ |
| M1 | With | $-732.5 \pm 11.3$ | $11.9 \pm 1.3$ | $99 \pm 12$ | $-703 \pm 52$ | 0.9952 | 0.9875 |
| | Without | $-726.9 \pm 11.1$ | $11.7 \pm 1.2$ | $87 \pm 10$ | $-702 \pm 41$ | 0.9967 | 0.9875 |
| M2 | With | $-784.8 \pm 9.9$ | $11.5 \pm 1.2$ | $115 \pm 15$ | $-574 \pm 33$ | 0.9961 | 0.9911 |
| | Without | $-763.2 \pm 9.5$ | $10.7 \pm 1.0$ | $95 \pm 9$ | $-574 \pm 25$ | 0.9976 | 0.9911 |
| M3 | With | $-751.6 \pm 10.7$ | $12.1 \pm 1.2$ | $111 \pm 9$ | $-492 \pm 39$ | 0.9973 | 0.9948 |
| | Without | $-740.4 \pm 10.2$ | $11.7 \pm 1.3$ | $97 \pm 10$ | $-492 \pm 41$ | 0.9985 | 0.9948 |

**Table 6.** Fitting results of polarization curves with and without an IR drop by Tafel extrapolation under three measurements in NaOH solutions after exposure for 2 h.

| Condtion | IR Drop | $E_{corr}$ mV | $i_{corr}$ μA·cm$^{-2}$ | $b_a$ mV·dec$^{-1}$ | $b_c$ mV·dec$^{-1}$ | Adj. R$^2$ | |
|---|---|---|---|---|---|---|---|
| | | | | | | $b_a$ | $b_c$ |
| M1 | With | $-270.3 \pm 9.6$ | $0.187 \pm 0.016$ | $275 \pm 12$ | $-167 \pm 7$ | 0.9974 | 0.9995 |
| | Without | $-274.5 \pm 9.2$ | $0.180 \pm 0.024$ | $275 \pm 13$ | $-166 \pm 11$ | 0.9974 | 0.9995 |
| M2 | With | $-268.2 \pm 9.8$ | $0.219 \pm 0.021$ | $261 \pm 19$ | $-183 \pm 12$ | 0.9989 | 0.9999 |
| | Without | $-268.6 \pm 9.0$ | $0.220 \pm 0.020$ | $261 \pm 17$ | $-183 \pm 9$ | 0.9989 | 0.9999 |
| M3 | With | $-277.1 \pm 9.1$ | $0.432 \pm 0.046$ | $291 \pm 19$ | $-222 \pm 15$ | 0.9971 | 0.9993 |
| | Without | $-277.1 \pm 8.6$ | $0.432 \pm 0.042$ | $291 \pm 12$ | $-222 \pm 16$ | 0.9971 | 0.9993 |

It can be seen from Figures 4 and 5 that there are obvious differences between the polarization curves with and without IR drops in the acidic solutions. The anodic curves

dramatically shift towards the negative direction, and the cathodic curves towards the positive direction, which indicates that the IR drop severely influences the shapes of anodic and cathodic curves and the fitting results. However, these kinds of differences are little in the neutral and alkaline solutions (Figures 6 and 7), which indicates that the IR drop has almost no effects on the polarization curves, especially for the cathodic curves. Table 2 shows that the solution resistances are smaller in the acid solution than those in the neutral and alkaline solutions. The effect of the IR drop on the polarization curve is less in the acidic solution than in the neutral or alkaline solution if the measured current densities are the same. However, as shown in Figure 2, it can be seen that there are much larger anodic and cathodic current densities in the strong polarizing regions in the acidic solutions. Thus, the effect of the IR drop on the polarization curve is much stronger in the acidic solutions. The results mentioned above imply that the IR drop has an important effect on the polarization curve in strongly corrosive systems due to the large polarizing current, even though the solution resistance is small. However, even if the solution resistance is big, the IR drop has few effects on the polarization curves if the coupons are placed in less corrosive electrolytes, e.g., NaOH solutions.

It can also be found from the fitting lines in Figures 4–7 that the IR drop severely influences the anodic and cathodic lines in the HCl and $H_2SO_4$ solutions, Figures 4 and 5, which indicates that wrong fitting data of anode and cathode will be obtained if the IR drop is not eliminated. Only anodic lines are changed by the IR drop in the NaCl solution, but there is almost not variation in the cathodic lines. For the NaOH solution, not only the anodic lines but also the cathodic lines remain uninfluenced.

As is well known, there are a large amount of hydrogen ions in the acidic solutions, which enhance both the anodic and cathodic reactions of steel coupons, i.e., the anodic and cathodic currents dramatically increase. As a result, IR drops can affect the anodic and cathodic curves. The NaCl solution can enhance the anodic reaction, but almost not at all for the cathodic reaction, which implies that only anodic curves are influenced by the IR drop. Similarly, the NaOH solution has few effects on the anodic and cathodic reactions, so the anodic and cathodic curves are not changed by the IR drop.

Tables 3–6 show that the fitting results for eliminating the solution resistance in acidic solutions differ significantly from those with an IR drop, but the fitting results with and without IR drop are essentially the same in neutral and alkaline solutions, which indicates that the IR drop seriously affects the fitting results of the polarization curves in acidic solutions, but almost not at all in neutral and alkaline solutions. It can also be seen that the Adj-$R^2$ values of the linear fits of the Tafel-polarization-heavy region for all three measurements are greater than 0.987, indicating that there are good linear relationships between the logarithmic values of the polarization potential and the polarization current density, i.e., the data selected from the measured polarization curves obey the Tafel–linear discipline well.

From Tables 3, 4 and 6, it can be seen that the anodic and cathodic Tafel slopes fitted by the three measurement methods in HCl, $H_2SO_4$, and NaOH solutions are similar, but that the corrosion current densities are very different, which indicates that these three methods have few effects on the anodic and cathodic reaction processes, and have huge effects on the fitted corrosion rates of coupons. The results in Table 5 show that the $b_c$ values calculated by the three measurement methods differ greatly in the NaCl solution, while the $b_a$ and $i_{corr}$ values are similar, which indicates that a change in measuring methods can influence the cathodic process of coupons in the NaCl solution. From the optical pictures (Figure S1), the corrosion products are not observed on the coupon surfaces in the HCl and $H_2SO_4$ solutions; however, there are plenty of corrosion products on the coupon surfaces in the NaCl and $Na_2SO_4$ solutions. It can be inferred that the difference in the fitted results might be related to the formation of corrosion products on the coupon surface, which will be discussed in a later paragraph.

It can be also seen from Tables 3 and 4 that the $b_a$ values are all more than 130 mV/dec which is far from 80 mV/dec in the presence of the IR drop in the HCl and $H_2SO_4$ solutions,

and that the $b_c$ values are much greater than 120 mV. After removing the effects of the ohmic resistance of solutions, the $b_a$ and $b_c$ values obviously decrease under three measuring conditions. The $b_a$ values of three measurements are in a range from 40 to 80 mV, and the $b_c$ values are closer to 120 mV/dec. Cao [20] has pointed out that the anodic Tafel slope ($b_a$) is usually from 30 to 120 mV/dec, and the cathodic Tafel slope ($b_c$) is more than 120 mV/dec. He has further concluded that the $b_a$ value is between 40 and 80 mV/dec and that the $b_c$ value is around 120 mV/dec in acid solutions under activation–dissolution conditions. The above results further imply that the IR drop can influence the accuracy of the fitting results of polarization curves.

### 4.2. Effects of Three Measuring Methods

As is well known, there is an electric double layer at a metal–electrolyte interface when the metal electrode is immersed in an electrolyte. Electrons are usually accumulated on the metal side, and positive-charged ions are on the solution side. The charge quantities of electrons and cations are the same when the electrode reaches equilibrium, as shown in Figure 8a. As the electrode is applied by a cathodic current or potential, as in Figure 8b, the electrons quickly accumulate on the side of the metal, and the cations in a solution begin to move towards the electrolyte side of the EDL under the effect of electrostatic attraction. The amounts of electrons are much larger than those of cations because of the faster migrating rate of electrons at the beginning of polarization. Only when re-reaching the equilibrium of the electrode do the quantities of positive and negative charges become the same on the both sides of the EDL.

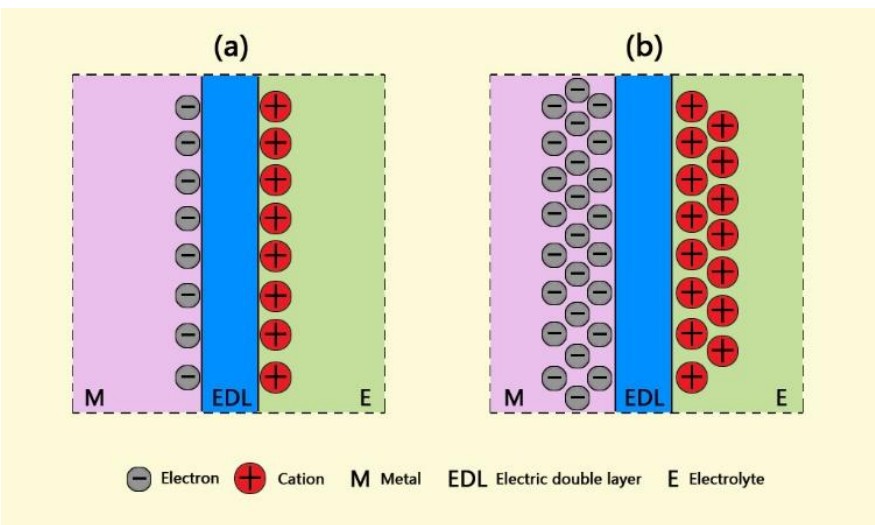

**Figure 8.** Diagram of an EDL with an equilibrium state of charges (**a**) and a state of applied cathodic current (**b**).

As shown in Figure 9a, the polarization currents measured ($i_M$) are divided into two parts according to Equation (12): faradaic current ($i_F$), which is used for the reaction of electrodes, and nonfaradaic current ($i_{NF}$), which is used for the charging of EDLs, called the charging current ($i_C$). When a small polarization potential is applied to the electrode, the equilibrium of charges between the sides of the EDL is easily reached; however, a large amount of electrons accumulate on the steel–solution interface if the applied potential is large enough, and there are not enough cations to quickly balance the negative charges, as shown in Figure 8b. In such cases, the discharging current ($i_{DC}$) is formed, and there is a relationship following Equation (13).

$$i_M = i_F + i_{NF} \tag{12}$$

$$i_M = i_F + (i_C - i_{DC}) \tag{13}$$

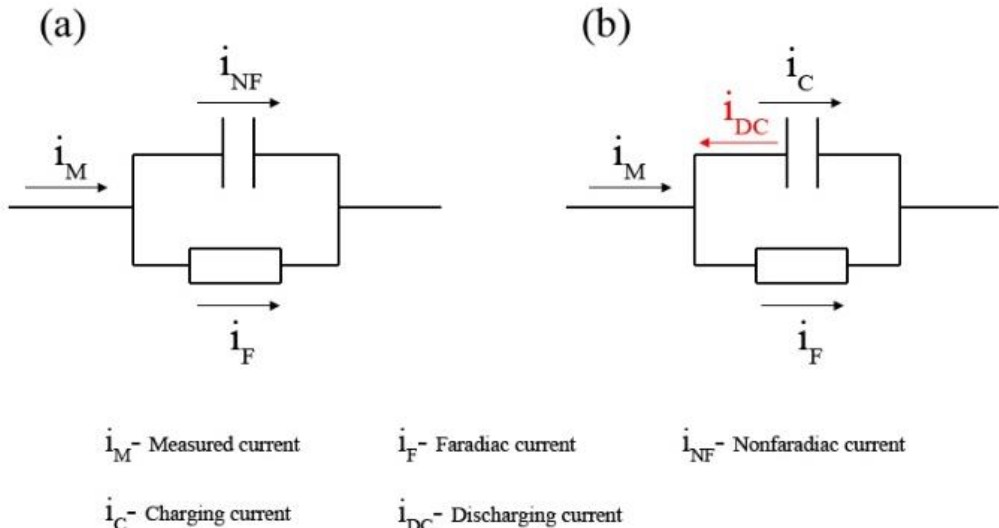

**Figure 9.** Diagram of equivalent electric circuits of electrode-applied currents (**a**) charging current, (**b**) the discharging current.

As is well know, supposing that the voltage applied to the EDL is constant, there is a positive relationship between the charge quantity of EDL and the capacitance, as described in Equation (14).

$$C = \frac{Q}{U} \tag{14}$$

For the condition of the polarization curve scanning from the OCP to the cathodic direction, the difference between the applied potential and the OCP is small at the beginning, which indicates that the increase in electrons on the steel side is not large, thus there are enough cations to quickly migrate to the solution side of the EDL and the equilibrium of charges is easily reached, as shown in Figure 8a. However, for the condition scanning from the cathodic direction to the anodic direction, with the application of a large voltage (e.g., −300 mV) to the steel coupon, a great quantity of electrons quickly accumulate on the steel side of the EDL, but the corresponding cations in the solution cannot migrate to the electrolyte side in time. As a result, an inverse current ($i_{DC}$) is formed, the value of which relates to the capacitance of the EDL when the applied potential is the same.

When corrosion products are not formed on the steel surface, e.g., in the acid solutions, as shown in Figure S1a,b, there is only a capacitance of the EDL of the interface. If the steel surface is covered by a layer of dense corrosion products or a passive film, as in Figure S1c–e, the total capacitance is the parallel capacitance of an EDL capacitance and a corrosion product (a passive film) capacitance. Studies [26,27] have shown that there is a layer of space charges in the corrosion products when a dense film of corrosion products is formed on the steel surface. This charge layer can be considered as a capacitance, the value of which is much smaller than that the EDL. In this case, the total capacitance value at the steel–solution interface is also smaller in the presence of corrosion products than that of only the EDL.

According to Equation (14), the charge quantities have a positive relationship with the capacitance when the voltage applied is constant. Supposing that either capacitances on the interface with or without corrosion products are considered as one capacitance. The charge quantities are much larger for the condition swithout corrosion products than those with the corrosion products, as the capacitance of the EDL is by far larger than that in the presence of corrosion products.

For the case scanning from −300 mV (vs. OCP), a large voltage is rapidly applied to the coupon at the beginning of polarization, i.e., this voltage is applied to the EDL at the interface, which leads to the accumulation of a large amount of electrons on the side of the metal. Corresponding cations are needed on the side of solution in order to reach an

equilibrium of charges. A reverse current, i.e., $i_{DC}$ in Figure 9b, is generated due to the reason that the migrating rate of an electron is much faster than that of a cation. Thus, larger current densities should be supplied for obtaining the set potential. Zhang et al. [26] have pointed out that the charging process of the capacitance of the interface usually influences the external current. Bard et al. [19] have also concluded that there is a flowing charging current because of the continuous change of the electrode potential during the potentiodynamic scanning process. The $i_{DC}$ value has a negative relationship with the capacitance, i.e., the smaller the capacitance value the larger the $i_{DC}$ value, which leads to an increase in $i_M$. Thus, there is a larger value of $i_{DC}$ when a layer of corrosion products is formed on the steel surface due to the smaller total capacitance. It can be seen from Figures 2 and S1 that there are horizontal lines for the M1 measurement in the cathodic curves in the neutral and alkaline solutions because the curves are scanned from the cathodic direction. Additionally, the lengths of the lines have a relationship with the properties of solutions. From Figure S1, the thickness of the corrosion product film is the largest in the $Na_2SO_4$ solution, and the length of the horizontal lines is also the longest. The thickness is the second in the NaCl solution, and the least in the NaOH solution.

As is well known, there is a negative relationship between the capacitance (C) and the layer thickness (d) as in the following Equation (15).

$$C = \varepsilon S/d \tag{15}$$

where C is a capacitance; $\varepsilon$ is a dielectric constant; d is a distance of the EDL and also a film thickness here.

Thus, it can be found that the capacitance value at the interface is the least in the $Na_2SO_4$ solution, and the largest in the NaOH solution.

Table 7 gives the maximum values of cathodic current densities in the neutral and alkaline solutions. The current density of M1 is larger than that of M2 or M3, and the current densities of M2 and M3 are similar. The current density of M1 is 131.3 $\mu A \cdot cm^{-2}$ (NaCl), 465.4 $\mu A \cdot cm^{-2}$ ($Na_2SO_4$), and 10.8 $\mu A \cdot cm^{-2}$ (NaOH) respectively, which is about 3.0, 7.5, and 1.4 times as much as those of M2 or M3. The maximum value of the cathodic current is much larger in the $Na_2SO_4$ solution than in the other two solutions, which leads to a huge change in the shape of the polarization curve, as shown in Figure 2d. The results mentioned above further verify the relationship between the maximum cathodic current density and the capacitance at the interface. The shape distortion will cause the wrong fitting results. As a result, the M1 measurement, i.e., scanning from the cathodic potential to anodic potential, is not suitable for the condition that a layer of corrosion products has been formed on the steel surface.

**Table 7.** Maximum values of cathodic current densities of polarization curves under three measuring methods in the neutral and alkaline solutions ($\mu A \cdot cm^{-2}$).

| Solution | NaCl | $Na_2SO_4$ | NaOH |
|---|---|---|---|
| M1 | 131.3 | 465.4 | 10.8 |
| M2 | 48.3 | 63.4 | 8.0 |
| M3 | 39.5 | 61.2 | 7.4 |

The time taken to re-reach the equilibrium of potential after the scanning finish of the cathodic curve by the M2 method is given in Table 8. The stabilization times of steel are less than 10 min in the acid solutions. However, the times are more than 30 min in the neutral and alkaline solutions, which might influence the measurements of the following anodic curves, especially for the neutral and alkaline solutions. Thus, the measuring method should be considered in order to obtain accurate results when the polarization curve is used.

**Table 8.** Balance time of $E_{OCP}$ for M2 after finishing the measurement of the cathodic curve.

|  | HCl | H$_2$SO$_4$ | NaCl | Na$_2$SO$_4$ | NaOH |
|---|---|---|---|---|---|
| Balance time of $E_{OCP}$ (min) | 6 | 9 | 32 | 30 | 35 |

## 5. Conclusions

In this study, the effects of the IR drop and scanning methods on Tafel curves are investigated in five solutions in order to find a method to solve the problems, and the results are as follows:

(1) The effect of the IR drop on Tafel curves is much greater in an acidic solution than in a neutral or an alkaline solution because there are larger polarizing currents in the acidic solution even though there is a small solution resistance. Thus, the IR drop value should be considered before the beginning of a polarization curve, especially for strong corrosive solutions.

(2) Although measuring programs designed by computers are suitable for some electrolytes and have few effects on the polarization curves, these fixed programs may induce some problems for others. Thus, it is better to select linear regions and linearly fit the data using related software for systems controlled by active reactions.

**Supplementary Materials:** The following supporting information can be downloaded at: https://www.mdpi.com/article/10.3390/coatings12091314/s1, Figure S1: Optical pictures in five solutions (HCl (a), H2SO4 (b), NaCl (c), Na2SO4 (d) and NaOH (e)) after immersing for 2 h, and macrographs of coupons taken out of the solutions in the red-color squares.

**Author Contributions:** Y.B. and J.X.: conceptualization, investigation, data curation, and writing—original draft. B.W.: supervision, investigation. C.S.: supervision, review, and editing. All authors have read and agreed to the published version of the manuscript.

**Funding:** This research was funded by the National Natural Science Foundation of China (Nos. 51771213 and 51871228).

**Institutional Review Board Statement:** Not applicable.

**Informed Consent Statement:** Not applicable.

**Data Availability Statement:** Data sharing is not applicable to this article.

**Acknowledgments:** The authors would like to thank Junhua Dong for his suggestions on this study and Ying Li for help with writing.

**Conflicts of Interest:** The authors declare no conflict of interest.

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
