# Peer review of "Effects of the IR Drop on the Electrochemical Corrosion of X80 Pipeline Steel in Different Solutions"

_coatings, doi:10.3390/coatings12091314_

Round 1

Reviewer 1 Report

This is anice and brief technical paper. I suggest to be published without any revision. 

Author Response

Dear dear reviewer

Re: Manuscript ID: coatings 1805473 and Title: Effects of IR Drop and Measurement Methods on Polarization Curves

Thank you for your letter and the reviewers’ comments concerning our manuscript entitled “Effects of IR Drop and Measurement Methods on Polarization Curves ” (1805473). l.Thanks to the reviewer for acknowledging this paper We would love to thank you for allowing us to resubmit a revised copy of the manuscript and we highly appreciate your time and consideration.

Sincerely

Yunlong Bai

Reviewer 2 Report

Comment 1: Title should be revised and improved.

Comment 2: Qualitative informations are missing in abstract. Abstract should be concise and the authors need to improve with more specific short results.

Comment 3: All acronyms should be introduced at first place of appearance in the text. For example IR. in the abstract section.

Comment 4: The introduction section should be modified and improved though citing recent references (2021 and 2022) related studies and indicating the novelty of the study compared to the carried works. Also, the following references should be included in the introduction part.

Comment 5: The purity of used products should be mentioned in the text.

Comment 6: The figures in the manuscript were poor, the author should improve the quality and solution of these figures.

Comment 7: Report the errors bars in tables 2, 3, 4, 5 and 6. The following references should added.

ü Journal of Colloid and Interface science. 574 (2020) 43-60. (https://doi.org/10.1016/j.jcis.2020.04.022).

ü Construction and Building Materials. 270 (2021) 121454. (https://doi.org/10.1016/j.conbuildmat.2020.121454),

ü Journal of Materials Research and Technology. 9 (2020) 2691-2703. (https://doi.org/10.1016/j.jmrt.2020.01.002)

ü Journal of Molecular Liquids 337 (2021) 116492 (https://doi.org/10.1016/j.molliq.2021.116492)

ü Journal of Molecular Liquids 322 (2021) 114549 (https://doi.org/10.1016/j.molliq.2020.114549)

Comment 8: Level of English is good however in a few places some syntax errors are present. At some places two or more words joined together that should be corrected.

Author Response

Dear dear reviewer

Re: Manuscript ID: coatings 1805473 and Title: Effects of IR Drop and Measurement Methods on Polarization Curves

  Thank you for your letter and the reviewers’ comments concerning our manuscript entitled “Effects of IR Drop and Measurement Methods on Polarization Curves ” (1805473). 

Comment 1: Title should be revised and improved.

Response 1: The title has been changed to “Effects of IR Drop on electrochemical corrosion of X80 pipeline steel in different solutions” (P1,L2)

Comment 2: Qualitative informations are missing in abstract. Abstract should be concise and the authors need to improve with more specific short results.

Response 2: It is supplemented in the abstract.”Three different test methods were used to test the polarization curve of X80 pipeline steel in five kinds of solutions. The different scanning direction lead to differences in cathodic polarization curves. In addition, the effect of IR drop on Tafel curve in acidic solution is significantly greater than that in neutral and alkaline solution”(P1,L15-18)

Comment 3: All acronyms should be introduced at first place of appearance in the text. For example IR. in the abstract section.

Response 3: To be revised in the manuscript , IR drop ( Deviation due to I (current) and R (resistance) ) (P1,L14)

Comment 4: The introduction section should be modified and improved though citing recent references (2021 and 2022) related studies and indicating the novelty of the study compared to the carried works. Also, the following references should be included in the introduction part.

Response 4: To be supplemented in the manuscript (P2,L63-70)

Comment 5: The purity of used products should be mentioned in the text.

Response 5: The reagents used in this study are all AR grades (P2,L92)

Comment 6: The figures in the manuscript were poor, the author should improve the quality and solution of these figures.

Response 6: The images in the manuscript were redrawn

Comment 7: Report the errors bars in tables 2, 3, 4, 5 and 6. The following references should added.

Response 7: The corresponding error values have been refined in tables 2, 3, 4, 5 and 6.

Comment 8: Level of English is good however in a few places some syntax errors are present. At some places two or more words joined together that should be corrected.

Response 8: To be revised in the manuscript.

Reviewer 3 Report

I read the manuscript entitled “Effects of IR Drop and Measurement Methods on Polarization Curves”. The subject is interesting because the analysis of the polarization curves is commonly quite complex. The results in this manuscript could be helpful to improve the analysis and to get better conclusions. However, the manuscript must be reviewed before its publications.

Below you will find my comments.

P# means page, and L# means line.

p3-L95: add a reference for EIS method.

Section: High-polarization region of Tafel extrapolation

P3: Equations must be reviewed. Missed the Faraday constant Please read your reference "Electrochemical methods fundamentals and applications by Allen J. Bard and Larry L. Faulker, page 100"
P3: Add reference for the equations
P3-L103: Equation 3 does not have "a0 and b0 constants".
P4-L: Use V or mV, but you should be consistent. See P4-L152 "-0.3 V vs. OCP" and P4-L156: "100 mV to 150 mV"

P6: Quality of figures is poor and must be improved to see the details on them.

P6-table 1, 3, 4, 5, and 6: ¿aode? ¿catode? something is wrong with these words.

P7: Equation 1 is repeated. See equation 11.

P8: Subscripts are not uniform. See “ba” in L281 and “ba”L288.

P18: Format of reference 2 and 6 is not uniform.

Author Response

Dear dear reviewer

Re: Manuscript ID: coatings 1805473 and Title: Effects of IR Drop and Measurement Methods on Polarization Curves Thank you for your letter and the reviewers’ comments concerning our manuscript entitled “Effects of IR Drop and Measurement Methods on Polarization Curves ” (1805473). We would love to thank you for allowing us to resubmit a revised copy of the manuscript and we highly appreciate your time and consideration.

p3-L95: add a reference for EIS method.

Response 1:add a reference, reference such as [18],P3,L107

Section: High-polarization region of Tafel extrapolation
P3: Equations must be reviewed. Missed the Faraday constant Please read your reference "Electrochemical methods fundamentals and applications by Allen J. Bard and Larry L. Faulker, page 100"  P3: Add reference for the equations

Response 2: Add a reference, reference such as [19], P3,L112

P3-L103: Equation 3 does not have "a0 and b0 constants".

Response 3: Corrections were made in the manuscript. P3,L114-116

P4-L: Use V or mV, but you should be consistent. See P4-L152 "-0.3 V vs. OCP" and P4-L156: "100 mV to 150 mV"

Response 4: The unit has been amended.|ΔE| from 0.1V to 0.15 V P4-L167

P6: Quality of figures is poor and must be improved to see the details on them.

P6-table 1, 3, 4, 5, and 6: ¿aode? ¿catode? something is wrong with these words.

Response 5: Table 1, 3, 4, 5, and 6 are modified in the manuscript,”anode & cathode”

P7: Equation 1 is repeated. See equation 11.

Response 6: The theory is the same, in order to correspond to the marks in Fig.3, it is redescribed.

P8: Subscripts are not uniform. See “ba” in L281 and “ba”L288.

Response 7: The subscript has been modified P8,L299

P18: Format of reference 2 and 6 is not uniform.

Response 8: The references were unified.

Sincerely.

Yunlong Bai

Round 2

Reviewer 3 Report

I read your manuscript again. It has been improved, but I have doubt about equations 3, 4, 5, 6, 7, and 8.

I read your reference: A J. Bard, L R. Faulkner, Electrochemical Methods- Fundamentals and Application, second ed., Wiley, New York, 2001. p. 233.

On page 233, I found the section: 6.2.4 Effect of Double-Layer Capacitance and Uncompensated Resistance. However, on page 100, equation 3.4.11, we have the butler-volmer equation. You can compare your equation 2 with equation 3.4.11. I just want to be sure that you have the butler-volmer equation well writted. From my point of view the faraday constant should be included in the mentioned equations.

Author Response

Point 1: On page 233, I found the section: 6.2.4 Effect of Double-Layer Capacitance and Uncompensated Resistance. However, on page 100, equation 3.4.11, we have the butler-volmer equation. You can compare your equation 2 with equation 3.4.11. I just want to be sure that you have the butler-volmer equation well writted. From my point of view the faraday constant should be included in

Response 1:The page number of the [19] references has been changed to P 100 according to the revision suggestion. And add F (faraday constantin )Butler-Volmer equation, which is respectively in equations 3,4,5,6,7 and 8
